# Spectral Differentiation of Hyperdense Non-Vascular and Vascular Renal Lesions Without Solid Components in Contrast-Enhanced Photon-Counting Detector CT Scans—A Pilot Study

**DOI:** 10.3390/diagnostics15010079

**Published:** 2025-01-01

**Authors:** Judith Becker, Laura-Marie Feitelson, Franka Risch, Luca Canalini, David Kaufmann, Ramona Wudy, Bertram Jehs, Mark Haerting, Claudia Wollny, Christian Scheurig-Muenkler, Thomas Kroencke, Florian Schwarz, Josua A. Decker, Stefanie Bette

**Affiliations:** 1Clinic for Diagnostic and Interventional Radiology and Neuroradiology, University Hospital Augsburg, Stenglinstr. 2, 86156 Augsburg, Germany; judith.becker@uk-augsburg.de (J.B.); laura-marie.feitelson@uk-augsburg.de (L.-M.F.); f.risch1@gmx.de (F.R.); lucanalini@gmail.com (L.C.); david.kaufmann@uk-augsburg.de (D.K.); ramona.wudy@uk-augsburg.de (R.W.); bertram.jehs@uk-augsburg.de (B.J.); mark-haerting@t-online.de (M.H.); claudia.wollny@uk-augsburg.de (C.W.); christian.scheurig@uk-augsburg.de (C.S.-M.); josua.decker@uk-augsburg.de (J.A.D.); stefanie.bette@uni-a.de (S.B.); 2Centre for Advanced Analytics and Predictive Sciences (CAAPS), University of Augsburg, Universitätsstr. 2, 86159 Augsburg, Germany; 3Centre for Diagnostic Imaging and Interventional Therapy, Donau-Isar-Klinikum, Perlasberger Straße 41, 94469 Deggendorf, Germany; florian.schwarz@donau-isar-klinikum.de

**Keywords:** renal lesions, vascular and non-vascular renal lesions, photon-counting detector CT, spectral decomposition, VNC, iodine quantification maps

## Abstract

**Introduction**: The number of incidental renal lesions identified in CT scans of the abdomen is increasing. Objective: The aim of this study was to determine whether hyperdense renal lesions without solid components in a portal venous CT scan can be clearly classified as vascular or non-vascular by material decomposition into iodine and water. **Methods:** This retrospective single-center study included 26 patients (mean age 72 years ± 9; 16 male) with 42 hyperdense renal lesions (>20 HU) in a contrast-enhanced Photon-Counting Detector CT scan (PCD-CT) between May and December 2022. Spectral decomposition into virtual non-contrast (VNC) images and iodine quantification maps was performed, and HU values were quantified within the lesions. Further imaging and histopathological reports served as reference standards. **Results:** Mean VNC values were 55.7 (±24.2) HU for non-vascular and 32.2 (±11.1) HU for vascular renal lesions. Mean values in the iodine maps were 5.7 (±7.8) HU for non-vascular and 33.3 (±19.0) HU for vascular renal lesions. Using a threshold of >20.3 HU in iodine maps, a total of 7/8 (87.5%) vascular lesions were correctly identified. **Conclusion:** This proof-of-principle study suggests that the routine use of spectral information acquired in PCD-CT scans might be able to reduce the necessary workup for hyperdense renal lesions without solid components. Further studies with larger patient cohorts are necessary to validate the results of this study and to determine the usefulness of this method in clinical routine.

## 1. Introduction

The use of computed tomography (CT) for clinical purposes, e.g., initial diagnosis or monitoring of cancer, has increased in recent years [1,2]. As a side effect, the number of incidental findings, especially of renal lesions, is increasing, leading to further examinations and burdening the health care system [3,4,5]. Homogeneous renal masses of −9 to 20 HU on contrast-enhanced CT scans of the abdomen or >70 HU on non-contrast CT can be classified as benign cysts [6]. On portal venous contrast-enhanced CT scans, the recommendations for classifying a homogeneous renal mass as benign vary from 20 to 30 HU [6,7]. However, the definition of some renal masses remains challenging on a single contrast CT scan and further examinations such as multiphase CT scans or MRI examinations are recommended to clarify [8,9,10]. Due to the limitations of widely used delayed-phase contrast CT scans only in standard imaging of the abdomen, it is not possible to distinguish between a true contrast enhancement and a hemorrhagic/proteinaceous cyst on initial imaging. Because minimally enhanced renal cell carcinomas without visually solid components may appear similar to a hemorrhagic/proteinaceous cyst, differentiation within a single contrast phase is difficult but of great importance. Studies have shown that dual-energy CT (DECT) scans, with the associated capability of virtual non-contrast (VNC) imaging and iodine uptake quantification maps, can identify incidental renal masses as non-vascular lesions (cysts with or without hemorrhagic/proteinaceous components) or vascular renal tumors [8,9,10]. However, this technique is not always performed routinely in clinical practice. Since 2021, a new generation of CT scanners has been available in clinical practice, using photon-counting detector (PCD) technology. With PCD-CT, it is possible to differentiate voxels into an iodine-attributable portion and into a non-iodine-attributable portion by exploiting the data’s spectral sensitivity [11]. In contrast to DECT, PCD-CT offers spectral differentiation within each scan, without higher radiation dose or special protocols. Therefore, VNC images and iodine quantification maps can be routinely reconstructed from a contrast-enhanced CT data set. Previous studies have demonstrated comparable values of VNC images to true non-contrast (TNC) images in dual-energy-CT scanners [12,13,14]. Studies have been performed to validate this statement for PCD-CTs. While some studies showed promising results, others showed restrained and improvable results [15,16,17]. The combination of non-contrast CT and contrast-enhanced CT scans clearly identifies the presence of contrast medium uptake in a renal mass. However, this requires an additional CT scan and increases the radiation exposure to the patient. While benign renal cysts do not show an enhancement on multiphase CT scans, renal cell carcinomas show measurable uptake [18,19]. If VNC images or iodine quantification maps are reliable and of diagnostic value in PCD-CT scans, TNC images or multiphasic CT scans may be unnecessary, leading to a reduction in radiation exposure and further investigations in initially indeterminate renal masses. Also, a delay in the final diagnosis and a potential psychological burden in patients due to uncertainty and further investigations might be avoided. Diagnostic workup for discriminating hyperdense renal cysts can be performed using either multiphasic CT, contrast-enhanced ultrasound (CEUS), MRI, PET-CT or histopathological diagnosis.

The aim of this study was to ascertain whether hyperdense renal masses (>20 HU) without apparent solid components on portal venous PCD-CT scans can be accurately classified as non-vascular (hemorrhagic/proteinaceous cysts) or vascular lesions (e.g., renal cell carcinoma) using VNC images and/or iodine quantification maps on a PCD-CT.

## 2. Materials and Methods

This retrospective single-center study was approved by the local Medical Research and Ethics Committee (Ludwig-Maximilians-University Munich, 23-0451) and the need for written informed consent was waived.

### 2.1. Study Population

Patients were retrospectively selected from our reporting system for contrast-enhanced CT examinations on our PCD-CT scanner with numerous keywords in our native language that best encapsulate the essence of our research, e.g., “renal cell carcinoma”, “increased density values”, “cystic renal lesion”, “increased HU”, “hyperdense” and “hemorrhagic/proteinaceous cyst/renal cyst”, from May 2021 to December 2022. Inclusion criteria were (1) full legal age (>18 years), (2) clinically indicated CT scan of the abdomen, (3) intravenous injection of contrast medium, (4) portal venous phase, (5) elevated HU (>20 HU) in a renal mass in a portal venous CT scan, and (6) available reference method (described in the section below). Exclusion criteria were (1) renal masses with obvious solid components, (2) size < 8 mm. Renal cell carcinomas were classified as vascular renal lesions, and hemorrhagic or proteinaceous renal cysts were classified as non-vascular renal lesions.

### 2.2. Scan Protocol and Reconstruction Settings

Contrast-enhanced CT scans were obtained as part of routine clinical care on a novel PCD-CT system (NAEOTOM Alpha, Siemens Healthineers, Erlangen, Germany). Patients were injected with 120 mL of contrast medium (Iopromide; Ultravist 300 mgI/mL, Bayer, Leverkusen, Germany) into an antecubital vein. This was followed by a 30 mL saline flush (flow rate: 4.0 mL/s). The scan started after bolus triggering with a delay of 45 s after 120 HU was reached in the ascending aorta. The scan direction was craniocaudal in a supine position from the lung apex or diaphragm to the symphysis in one single breath-hold. We used an acquisition mode with readout of spectral information (QuantumPlus, Siemens Healthineers, Forchheim, Germany), a tube voltage of 120 kVp, automatic tube current modulation (Care DOSE 4D, Siemens Healthineers), a rotation time of 0.25 s, a pitch of 0.8 and a collimation of 144 × 0.4 mm^2^. Spectral series were reconstructed with a soft tissue kernel (Qr40, QIR 3, Siemens Healthineers). All images contained spectral information (SPP, spectral postprocessing, Siemens Healthineers). A slice thickness of 1.0 mm and an increment of 1.0 mm were applied.

### 2.3. Image Analysis

Quantitative analyses were performed using a dedicated workstation called Syn-go.Via (VB60A, Siemens Healthineers, Erlangen, Germany) and the picture archiving and communicating system Deep Unity (Dedalus Health Care, Bonn, Germany). Two radiology residents and one board-certified radiologist (4, 6 and 10 years’ experience) manually placed regions of interest (ROIs) in the conspicuous renal masses using Deep Unity as well as Syngo.Via. ROIs of identical size were placed in the renal masses. All evaluations were performed on axial reconstructions and in the portal venous phase. For all patients’ iodine quantification maps, VNC images and virtual monoenergetic reconstructions (VMI, 70 keV) were performed using Syngo.Via and CT values were quantified within the lesions. Three measurements were obtained within each renal mass and the mean value was used for further calculations. If the calculated iodine attenuation was negative in a renal lesion, it was set to “0” for further calculation, assuming that there was no negative iodine uptake (*n* = 12) [20].

### 2.4. Reference Standard

Renal lesions were clearly classified as non-vascular or vascular lesions if (1) a TNC scan was available, (2) ultrasound or CEUS were performed, (3) an MRI or PET-CT examination was present or (4) in case of a histopathologic report either by biopsy or surgery.

### 2.5. Statistical Analysis

Data were analyzed using R (R Statistics, version 4.3.1, R Core Team, Vienna, Aus-tria) [21] and RStudio (version 2023.06.2) [22]. To assess normal distribution of data, the Shapiro–Wilk test was performed. Data are presented as median with interquartile range (IQR) or as mean ± standard deviation as indicated. Non-normally distributed continuous data were compared using the Mann–Whitney U test and normally distributed data were compared using *t*-tests. To select the best discriminating features, the random forest model Boruta was applied in R. The best cutoff values for discriminating between vascular and non-vascular renal lesions were calculated in R. Statistically significant differences were assumed at *p*-values ≤ 0.05.

## 3. Results

### 3.1. Patient Baseline Characteristics

In total, 110 renal masses of increased density values (>20 HU) were identified. In total, 68 renal lesions had to be excluded because no further clarification of the renal lesions was performed. Four lesions had to be excluded because the size was too small for further measurements (<8 mm). Four lesions were excluded because of a missing portal venous phase and fifty lesions due to missing further work-up. Ten lesions were excluded because the renal mass could be clearly attributed to a vascular lesion because of obvious solid components (Figure 1).

Finally, a total of 26 patients (mean age 72 years ± 9, 16 assigned male at birth) with 42 ascertained renal lesions were included in this study. Median BMI was 26.8 kg/m^2^ (interquartile range [IQR] 25.5–27.5 kg/m^2^). Baseline characteristics are shown in Table 1 (Table 1). The entity of the renal masses was confirmed based on pathology reports or additional imaging exams (ultrasound, CEUS, TNC images, multiphasic CT scan, MRI and/or PET-CT) (Table 2).

### 3.2. Quantitative Image Analysis

Using the standard VMI reconstruction of abdominal CT scans (70 keV), there was a clear overlap of the measured HU in non-vascular and vascular renal lesions on contrast-enhanced CT scans in the portal venous phase. The mean CT values for non-vascular lesions were 61.4 (±19.4) HU, while for vascular lesions, the mean CT values were 65.5 (±17.4) HU (*p*-value = 1.000) (Figure 2). The mean VNC values differed significantly between vascular renal lesions (32.2 ± 11.1 HU) and non-vascular renal lesions (55.7 ± 24.2 HU) (*p*-value = 0.034). Significant differences were also observed in the acquired iodine quantification maps of non-vascular renal lesions (mean 5.7 ± 7.8 HU) and vascular renal lesions (33.3 ± 19.0 HU) (*p*-value = 0.002) (Table 3).

Figure 3 illustrates two cases of hyperdense renal masses without solid components, which exhibited increased HU in a portal venous CT scan (75/86 HU), leading to an uncertainty in diagnosis. These lesions required further examinations. Using CEUS, these lesions were identified as a cyst in one patient because no contrast medium uptake was present (A) and as a vascular renal lesion, suspicious for renal cell carcinoma, in the other patient based on the contrast medium uptake, which was later confirmed by histopathology (B). Interestingly, in this case, the renal cell carcinoma presented completely without solid components, which aggravates the diagnosis, especially in single-contrast CT examinations, and further examinations are required (Figure 3).

As the best cut-off value for discrimination between vascular and non-vascular renal lesions in iodine quantification maps, 20.3 HU was identified, with a sensitivity and specificity of 87.5% and 88.2%. For VNC images, 52 HU was shown as the best cut-off value, also with a sensitivity of 100.0% but with a lower specificity (55.9%). Additionally, 70 keV reconstructions also showed lower sensitivity (76.5%) but still a specificity of 50% for discrimination when using a cut-off value of 74 HU (Table 4).

Using iodine quantification maps, a total of 7/8 (87.5%) vascular lesions were correctly identified.

Using a random forest model for feature selection, iodine quantification maps were identified as the most important features for differentiation between vascular and non-vascular renal lesions without solid components (Figure 4, Table 5). Also, VNC and 70 keV VMI were confirmed as important features, with, however, lower importance values compared to iodine quantification maps.

## 4. Discussion

In this study, we investigated the potential of spectral differentiation in VNC images and iodine quantification maps to distinguish between hyperdense non-vascular and vascular renal lesions without solid components in PCD-CT scans in a portal venous phase. The main results were as follows: (1) HU values in vascular renal lesions and non-vascular hyperdense renal lesions clearly overlap in VMI reconstructions (70 keV), and therefore it is impossible to distinguish between those two entities in the standard reconstructions which requires further examinations; (2) VNC values are able to differ between non-vascular renal lesions and vascular renal lesions using a cut-off value of 52 HU (sensitivity 100%, specificity 55.9%); and (3) iodine quantification maps might be able to identify non-vascular renal lesions from vascular renal lesions using a cut-off value of 20.3 HU.

As the number of incidental renal lesions is constantly increasing, classification in initial imaging is of great importance to reduce the burden of follow-up examinations, delays in diagnosis, psychological burden on the patients and unnecessary health care costs [3,4,5]. Although the vast majority of incidental renal lesions are benign, most renal cell carcinomas are detected incidentally [23]. Renal cell carcinomas account for around 3% of all cancers, and men are more frequently affected than women [24]. In recent years, their incidence has increased [24]. Detecting small renal cell carcinomas is of absolute importance, as surgery remains the only curative treatment at an early stage. It is essential not to overlook these types of carcinomas in incidental renal masses identified during imaging [25]. Computed tomography, magnetic resonance imaging and sonography are the main methods used to characterize renal lesions, but sometimes a combination of these imaging modalities is necessary to characterize a renal mass with certainty [26]. For the differentiation of cystic renal lesions into benign or malignant, the Bosniak Classification for contrast-enhanced CT scans was developed and updated in 2019 [27]. This classification helps in standardizing the interpretation of renal masses and therefore facilitates decision making for further follow-up or surgical treatment. However, it remains difficult to distinguish a hemorrhagic/proteinaceous renal cyst from a vascular renal mass without visible solid components in a portal venous contrast-enhanced CT scan. Renal cell carcinomas with necrotic parts complicate diagnosis, although these are quite rare [28]. To distinguish between a hemorrhagic/proteinaceous cyst and a real contrast enhancement in vascular renal tumors, an additional TNC scan or a multiphasic CT scan are required. However, this increases the patients’ radiation exposure and therefore should be carefully considered [29,30,31]. Due to the indication of further examinations, the final diagnosis might be delayed, which also might lead to a psychological burden for the patients. Therefore, it is of high importance to find alternatives that speed up the diagnosis and avoid further tests or examinations.

Since 2021, PCD-CT scans have been available in clinical routine. In comparison to energy-integrating detector (EID) CT, PCD-CT directly converts incoming photons into electric signals and convinces with lower radiation doses and increased resolution [11,16]. With PCD-CT, spectral material decomposition is possible (similar to DECT). The main advantage of PCD-CT is the possibility to perform further reconstructions (i.e., VNC, iodine quantification maps) after each scan and without applying specific protocols [11]. However, many centers using fast-switching and dual-source DECT also routinely acquire spectral data on abdominal CT for characterizing incidental lesions, and thus have the spectral data available.

This study highlights that by using iodine quantification maps, routinely obtained from PCD-CT scans, hyperdense non-vascular renal lesions can be distinguished from vascular renal masses without obvious solid components in a total of 7/8 cases (87.5%). The reconstruction of iodine quantification maps and VNC images is also achievable using DECT scans, demonstrating high reliability [12,13,14,32]. Previous studies on DECT also revealed promising results for iodine quantification maps to prove a contrast medium enhancement in renal masses and therefore to differentiate vascular from non-vascular renal lesions [8,32,33,34]. Mastrodicasa et al. confirmed that vascular renal lesions can be significantly differentiated from hemorrhagic/proteinaceous cysts using iodine quantification maps in DECT [8]. The main advantage of PCD-CT compared to DECT is the fact that spectral data are acquired routinely. No further default settings have to be implemented and the necessary reconstructions can be carried out at any time after the examination if required. Especially in incidentally detected lesions, this is very important.

The use of VNC images showed no significant findings for differentiating vascular renal lesions from hemorrhagic/proteinaceous renal masses [8]. This conclusion can also be transferred to our study with PCD-CT scans. In the present study, the HU values in VNC images of hemorrhagic/proteinaceous renal lesions and vascular tumors clearly overlap, leading to a differentiation of these entities with a sensitivity of 100% and a specificity of 55.9%. Although recent PCD-CT studies have shown the reliability of VNC reconstructions, some studies have highlighted the need for further improvement due to over- and underestimation compared to TNC images [17,35,36,37,38,39,40].

Based on these critical studies and the results of this study, the value of VNC reconstructions for differentiating hyperdense non-vascular from vascular renal lesions in clinical routine is limited and cannot be used with absolute certainty in clinical diagnostics. Similar results were shown in a previous study that analyzed the value of VNC images in differentiating renal masses in DECT. Verstraeten et al. showed that by using VNC images, incidentally detected renal lesions can be accurately characterized, but the largest difference was found for unenhanced hyperdense renal lesions, underlining the above-mentioned problem of characterizing hyperdense non-vascular renal lesions [41]. Bucolo et al., on the other hand, concluded that VNC and TNC images have a strong agreement, with a small remaining preference for TNC images in characterization of renal lesions [42].

Cut-off values for classifying hyperdense renal masses in portal venous CT scans of the abdomen differ between 20 and 30 HU [6,43]. In the study by Mastrodicasa et al., a cut-off value of 31.55 HU could differentiate vascular renal lesions from hemorrhagic/proteinaceous cysts with a sensitivity of 96% and a specificity of 10% [8]. In this study, the optimal threshold in the 70 keV VMI reconstructions for differentiating those two entities was 74.0 HU (76.5% sensitivity and 50.0% specificity), and therefore much higher in comparison. For iodine quantification maps, Mastrodicasa et al. measured the optimal cut-off values in iodine concentrations [8]. To our knowledge, this is the first study to use a cut-off value in iodine quantification maps given in HU for PCD-CT, assuming to facilitate decision making due to a commonly used unit. With a cut-off value of 20.3 HU in iodine quantification maps, vascular renal masses can be differentiated from proteinaceous/hemorrhagic cysts with >87.5% sensitivity and specificity on a PCD-CT. Since this is the first study on a PCD-CT to suggest such a cut-off value in HU, further studies will be necessary to verify these results. There have been previous studies on DECT and spectral-detector CT (SDCT) that also analyzed iodine quantification maps and iodine–water maps for differentiating renal lesions. All studies pointed out the value of iodine maps for discriminating vascular and non-vascular renal lesions [32,44,45].

It is important to note that the chosen energy levels in VMI reconstructions have a significant impact on image evaluation [46]. Both DECT and PCD-CT allow the reconstruction of various energy levels for the examination of CT scans [11,47]. In earlier CT devices, it was not possible to visualize these energy levels and conventional polychromatic 120-kVp images were routinely performed in abdominal CT scans [48]. Recent studies have compared VMI at 70 keV with polychromatic 120 kVp images; these studies have concluded that the VMI reconstructions at 70 keV exhibit superior image quality [48,49]. VMI at 70 keV provided reliable and reproducible results in the assessment of renal lesions, as well [50,51]. An advantage of evaluating renal masses at 70 keV is the lower occurrence of pseudoenhancement [50,51]. If there is a contrast difference of >20 HU between native and contrast-enhanced sequences in CT scans, a clear contrast medium uptake is assumed; if an enhancement of less than 20 HU is present, this might be due to pseudoenhancement [52]. The main discussed reason for the phenomenon of pseudoenhancement is due to beam-hardening effects [53]. In this study, we used VMI reconstructions at 70 keV as well for characterizing renal masses to account for these findings.

The strength of this study is the inclusion of hemorrhagic/proteinaceous renal cysts and renal carcinomas without obvious solid components only; classification of the latter entity is the greatest problem in imaging, but of absolute importance due to the need for further therapy. A delay of the final diagnosis due to further necessary examinations might play a vital role in treatment, as these malignant tumors can grow within a short period of time and can only be cured in an early stage of cancer.

This study has limitations. The main limitation is the small number of patients with vascular renal lesions without solid components. This entity is very rare, but remains an important differential diagnosis. This study only included cystic renal lesions without obvious solid components; most malignant tumors show both cystic and solid lesions and are in most cases easy to differentiate from benign lesions. However, the major challenge is to discriminate completely cystic malignant lesions from benign hyperdense cystic lesions. Therefore, this study only included lesions without solid components, although this led to a small number of malignant lesions. Due to the low number of vascular lesions, no subgroup analyses regarding the histopathology (e.g., clear cell vs. papillary renal cell carcinoma) were feasible. However, these might present with different contrast enhancements and might be considered in larger study cohorts. Further studies (e.g., multicenter studies, prospective studies or studies with a unique and standardized gold standard) with a larger patient collective are necessary to confirm the results of this study and to verify the obtained results in terms of validity and reproducibility. Moreover, a comparison between the diagnostic accuracy of DECT and PCD-CT for hyperdense cystic renal lesions would be of high importance.

In this study, we retrospectively selected hyperdense (>20 HU) cystic renal lesions with available contrast-enhanced PCD-CT. A large portion of patients were excluded due to missing follow-up or missing workup of the lesions. This selection of abnormal cases might introduce an unavoidable bias and needs to be considered in future studies.

## 5. Conclusions

Iodine quantification maps on PCD-CT scans may be able to reliably differentiate hyperdense non-vascular renal lesions from vascular renal lesions without obvious solid components, exhibiting an excellent sensitivity and specificity of >87%. Consequently, additional follow-up examinations and TNC CT scans, leading to increased radiation exposure and delayed diagnosis, may be unnecessary, thereby contributing to a reduction in radiation dose and health care system costs. Further studies with larger patient cohorts are necessary to prove the results of this pilot study.

## Figures and Tables

**Figure 1 diagnostics-15-00079-f001:**
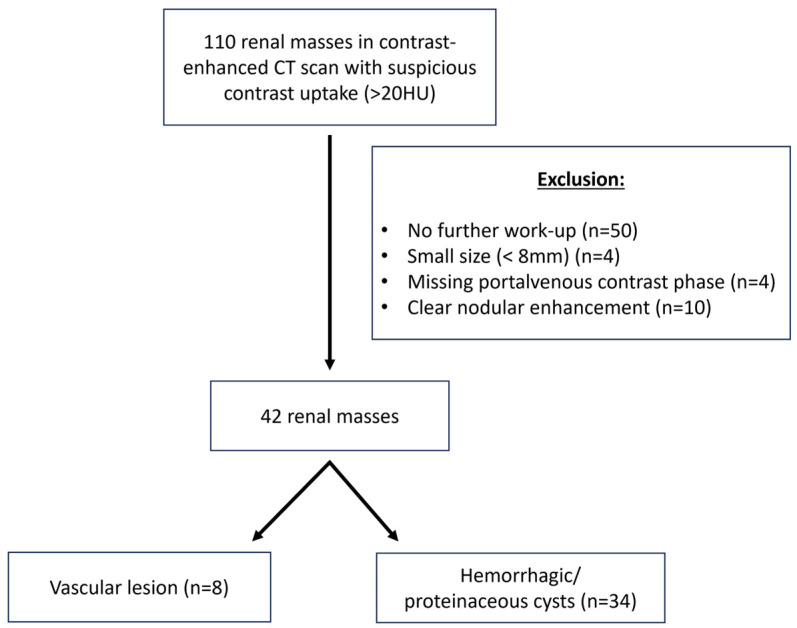
Flowchart of renal masses based on exclusion/inclusion criteria.

**Figure 2 diagnostics-15-00079-f002:**
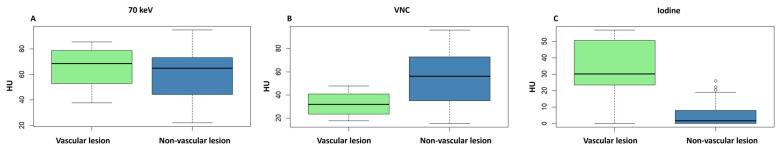
Boxplots for HU values in iodine quantification maps (**A**), virtual non-contrast (VNC) reconstructions (**B**) and 70 keV virtual monoenergetic reconstructions (VMI) (**C**).

**Figure 3 diagnostics-15-00079-f003:**
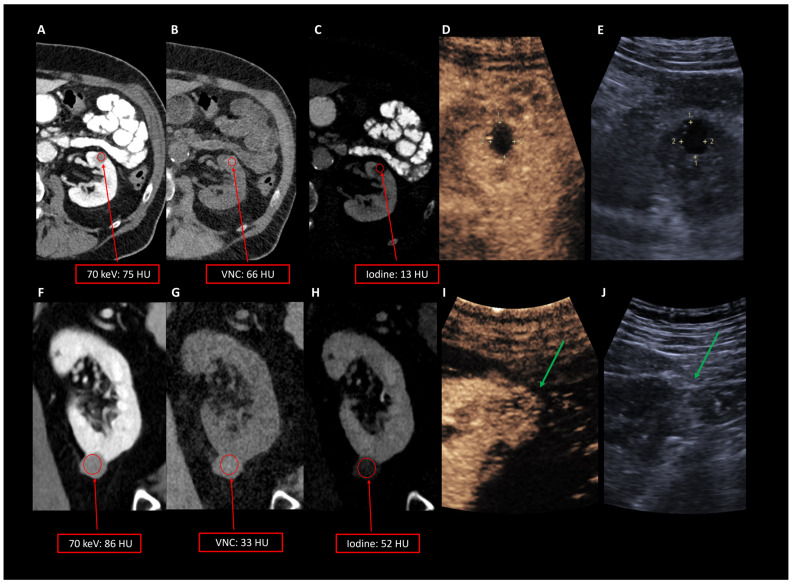
(**A**): Axial 70 keV reconstruction as well as VNC and iodine maps of a left renal mass in a portal venous CT scan of the abdomen. The lesion shows elevated HU in 70 keV (75 HU), VNC values of 66 HU (**B**) and 13 HU for iodine map CT values (**C**). (**D**,**E**): Contrast-enhanced ultrasound confirmed that there was no contrast medium uptake. (**F**–**H**): Example of a renal mass without solid components and elevated CT values in iodine maps in comparison to the first patient (52 HU) and similar values in 70 keV reconstructions (86 HU) compared to the confirmed renal cyst (**A**). Contrast-enhanced ultrasound confirmed a contrast enhancement (green arrow in (**I**,**J**)) and histopathology revealed the diagnosis of a renal cell carcinoma.

**Figure 4 diagnostics-15-00079-f004:**
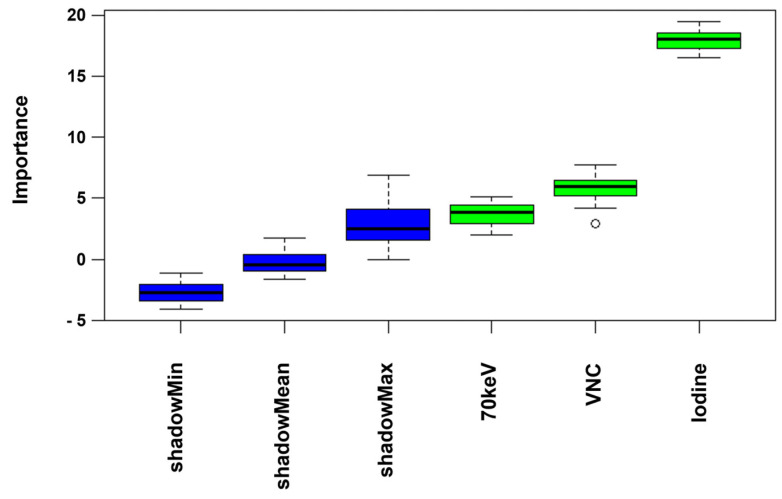
Feature importance in a random forest model (Boruta).

**Table 1 diagnostics-15-00079-t001:** Baseline patient characteristics.

Age (Years), Mean (±sd)	72.0 Years (±9.0)
Male, *n* (%)	16/26 (61.5%)
BMI [kg/m^2^], median [IQR]	26.8 (25.5–27.5)
Mean CTDI_vol_ [mGy] (±sd)	8.7 (±2.99)
Number of lesions	42
Non-vascular, *n* (%) Vascular, *n* (%)	34/42 (81.0%) 8/42 (19.0%)
Maximum diameter, mean (±sd)	20.0 mm (±8.0)

Values are mean ± standard deviation or median and interquartile range. BMI = body mass index, CT = computed tomography, CTDI_vol_ = computed tomography dose index.

**Table 2 diagnostics-15-00079-t002:** Reference methods.

Reference Methods	*n*
Follow-up (>6 months)	16
Multiphasic or TNC CT scan	5
MRI	2
B-mode Ultrasound	5
CEUS	6
Biopsy	2
PET-CT	6

TNC = true non-contrast, CT = computed tomography, MRI = magnetic resonance imaging, CEUS = contrast-enhanced ultrasound.

**Table 3 diagnostics-15-00079-t003:** CT values for virtual monoenergetic reconstructions (70 keV and VNC) and iodine maps.

	Non-Vascular	Vascular	*p*-Value
70 keV	64.8 (44.5–73.1) 61.4 (±19.4)	68.5 (58.0–78.5) 65.5 (±17.4)	1.000
Iodine maps	56.2 (35.6–72.7) 55.7 (±24.2)	31.8 (25.8–38.3) 32.2 (±11.1)	0.034
VNC	1.7 (0.0–8.0) 5.7 (±7.8)	30.3 (25.1–49.0) 33.3 (±19.0)	0.002

Data are shown as median (interquartile range) and mean ± standard deviation; *p*-values from Mann–Whitney U tests (iodine quantification maps) and *t*-tests (VNC and 70 keV).

**Table 4 diagnostics-15-00079-t004:** Best cut-off values.

Threshold (HU)	Specificity	Sensitivity	Accuracy
Iodine maps			
20.3	0.882	0.875	0.881
VNC			
52.0	0.559	1.000	0.643
70 keV			
74.0	0.500	0.765	0.714

VNC: virtual non-contrast.

**Table 5 diagnostics-15-00079-t005:** Feature importance.

	Mean Importance	Median Importance	Minimum Importance	Maximum Importance	Decision
70 keV	3.69	3.87	1.99	5.08	Confirmed
VNC	5.84	5.96	2.91	7.72	Confirmed
Iodine Maps	18.01	18.08	16.54	19.49	Confirmed

VNC: virtual non-contrast.

## Data Availability

The data presented in this study are available on request from the corresponding author.

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
