# Peer review of "Spectral Differentiation of Hyperdense Non-Vascular and Vascular Renal Lesions Without Solid Components in Contrast-Enhanced Photon-Counting Detector CT Scans—A Pilot Study"

_diagnostics, 2025, doi:10.3390/diagnostics15010079_

Round 1

Reviewer 1 Report (Previous Reviewer 2)

Comments and Suggestions for Authors

Thank you for your reply.

Author Response

We thank the Reviewer again for consideration of our manuscript and the positive view of the revised version of the manuscript.

Reviewer 2 Report (New Reviewer)

Comments and Suggestions for Authors

Dear authors,

The journal article is exceptionally well-written, providing a comprehensive and insightful examination of highly advanced photon-counting CT (PCCT) imaging. It effectively addresses the cutting-edge developments in this rapidly evolving field. The conclusion of the study highlights the promising potential of iodine quantification maps derived from photon-counting detector (PCD) CT scans in differentiating hyperdense non-vascular renal lesions from vascular ones. I hope that you will perform further research with larger patient cohorts to validate these results, underscoring the potential of photon-counting CT as an effective and efficient diagnostic approach in clinical practice.

Reviewer

Author Response

Thank you very much for considering our manuscript and for the positive view of the study. We will continue research in the field of photon-counting CT with the aim to further apply this new technology and the potential of spectral differentiation in clinical routine.

Reviewer 3 Report (New Reviewer)

Comments and Suggestions for Authors

Becker et al selected CT images reporting abnormal findings of renal cysts/hyperdense lesion/increased HU/RCC etc and investigated 26 scans for the sensitivity and specificity of contrast enhanced photon counting detector CT scanning to differentiate between vascular and non-vascular lesions.

I have the following comments:

 - please add the study characteristics to the abstract: mono centric, retrospective analysis to ease understanding at first reading

 - I suggest to call it a proof of principle study to elucidate the possibility of using PCD-CT in early cancer diagnostics as the cohort is too small to make any conclusion and state in the abstract that further analysis is necessary to determine the usefulness given the low sensitivity/specificity 

- the major issue is the study design as the authors selected images according to abnormal reporting instead of a more unbiased approach to investigate all images performed, one does not know whether PCD-CT would have detected lesions not highlighted in routine scanning, I suggest to add this to the limitations, 50 scans 'with no further work-up' were excluded, would the authors please elaborate on this

- 42 masses were selected in the study flow chart but only 26 patients investigated, what happened to the residual 16 scans

- a further major issue is the statistics applied, an initial cohort is investigated without validation and no second cohort used, the coarse ROC curves are the result of the small number investigated and the calculations are too vulnerable to use and subject to overfitting, the perfect AUC=1 is misleading due to the construct and small n, I suggest to refrain from attempting any performance analysis and to just highlight the percentages/numbers where PCD scans identify the vascular vs non-vascular accurately as an initial experiment requiring further studies

Author Response

Thank you very much for considering our manuscript and for the valuable comments that were addressed in the revised version of the manuscript.

I have the following comments:

 - please add the study characteristics to the abstract: mono centric, retrospective analysis to ease understanding at first reading

Thank you for this suggestion; we included the study characteristics in the abstract.

 - I suggest to call it a proof of principle study to elucidate the possibility of using PCD-CT in early cancer diagnostics as the cohort is too small to make any conclusion and state in the abstract that further analysis is necessary to determine the usefulness given the low sensitivity/specificity

Thank you for this comment; we added this in the abstract and changed the last sentences in the abstract accordingly:

“This proof of principle study suggests that routine utilization of spectral information acquired in PCD-CT scans might be able to reduce the necessary workup for hyperdense renal lesions without solid components. Further study with larger patient cohorts are necessary to validate the results of this study and to determine the usefulness of this method in clinical routine.“

- the major issue is the study design as the authors selected images according to abnormal reporting instead of a more unbiased approach to investigate all images performed, one does not know whether PCD-CT would have detected lesions not highlighted in routine scanning, I suggest to add this to the limitations, 50 scans 'with no further work-up' were excluded, would the authors please elaborate on this

Thank you for these valuable comments; we added a new paragraph in the limitations section:

“In this study, we retrospectively selected hyperdense (>20 HU) cystic renal lesions with available contrast-enhanced PCD-CT. A large portion of patients was excluded due to missing follow-up or missing workup of the lesions. This selection of abnormal cases might introduce an unavoidable bias and needs to be considered in future studies.”

- 42 masses were selected in the study flow chart but only 26 patients investigated, what happened to the residual 16 scans

Thank you for this comment. This is because more than one lesion was included per patient. This study includes 26 patients with 42 hyperdense cystic renal lesions. We clarified this in the first paragraph of the results section.

- a further major issue is the statistics applied, an initial cohort is investigated without validation and no second cohort used, the coarse ROC curves are the result of the small number investigated and the calculations are too vulnerable to use and subject to overfitting, the perfect AUC=1 is misleading due to the construct and small n, I suggest to refrain from attempting any performance analysis and to just highlight the percentages/numbers where PCD scans identify the vascular vs non-vascular accurately as an initial experiment requiring further studies.

Thank you for these valuable comments. We totally agree with you that ROC curves might be prone to overfitting in this small cohort. As suggested, we deleted this analysis and only highlighted the number of lesions that were accurately identified via PCD-CT iodine quantification maps (7/8 renal lesions).

Round 2

Reviewer 3 Report (New Reviewer)

Comments and Suggestions for Authors

The authors have responded to my queries, I do not have any further comments.

This manuscript is a resubmission of an earlier submission. The following is a list of the peer review reports and author responses from that submission.

Round 1

Reviewer 1 Report

Comments and Suggestions for Authors

What is the intent of the study? The article is a clinical study about the role of Photon-Counting CT in differentiation between vascular and non-vascular renal lesions without solid components. The authors focus on the role of the iodine quantification maps, the virtual non-contrast images and the 70 Kev virtual monoenergetic reconstructions to differentiate these two different types of renal lesions.

What conclusions do the authors reach? VNC values and iodine quantification maps are able to differ between vascular and non-vascular renal lesions with an high specificity and sensitivity.

Author Response

Spectral differentiation of hyperdense non-vascular and vascular renal lesions without solid components in contrast-enhanced Photon-Counting Detector CT scans – a pilot study (diagnostics-2774666)

We would like to thank the editor-in-chief and the reviewers for their effort and highly appreciate the constructive character of their criticism and comments. The manuscript was revised as suggested and resubmitted with the following point by point response to the comments of the reviewers. We believe that the quality of our manuscript substantially improved during the review process.

Response to Reviewer 1:

What is the intent of the study? The article is a clinical study about the role of Photon-Counting CT in differentiation between vascular and non-vascular renal lesions without solid components. The authors focus on the role of the iodine quantification maps, the virtual non-contrast images and the 70 Kev virtual monoenergetic reconstructions to differentiate these two different types of renal lesions.

What conclusions do the authors reach? VNC values and iodine quantification maps are able to differ between vascular and non-vascular renal lesions with an high specificity and sensitivity.

We would like to thank the reviewer for the time and consideration of our manuscript. 

Reviewer 2 Report

Comments and Suggestions for Authors

The cohorts of patients with vascular (n=4) and nonvascular renal lesions (n=31) were highly imbalanced, and I don't think that any meaningful and generalizable conclusions can be drawn with as few as 4 vascular lesions.

Moreover, vascular renal lesions can differ based on their histology, leading to markedly different contrast enhancement patterns. A typical example is given by clear cell vs papillary renal carcinomas - the latter can be much harder to detect due to their faint enhancement, whereas clear cell RCC are usually easily detected as avidly hypervascular masses. Hence, it is important to specify which histotype of renal cancer was put under the 'vascular lesions' umbrella, and eventually do a subclass analysis of different histotypes - unless one decides apriori to enroll e.g. ccRCCs only.

Finally, it would be important to have a control group of patients imaged with a non-PCCT scanner (ideally a DECT, EID-based type) to seek differences in diagnostic performance. The advantages of DECT over single energy CT are well known since the early days of abdominal DECT, and comparing the study findings with non-PCCT DECT instead of presenting them without an external comparison would add considerable value to the manuscript.

Comments on the Quality of English Language

A minor editing of the English language should be performed.

Author Response

Spectral differentiation of hyperdense non-vascular and vascular renal lesions without solid components in contrast-enhanced Photon-Counting Detector CT scans – a pilot study (diagnostics-2774666)

We would like to thank the editor-in-chief and the reviewers for their effort and highly appreciate the constructive character of their criticism and comments. The manuscript was revised as suggested and resubmitted with the following point by point response to the comments of the reviewers. We believe that the quality of our manuscript substantially improved during the review process.

Response to Reviewer 2:

The cohorts of patients with vascular (n=4) and nonvascular renal lesions (n=31) were highly imbalanced, and I don't think that any meaningful and generalizable conclusions can be drawn with as few as 4 vascular lesions.

We thank the reviewer for consideration of our manuscript and for providing valuable comments. We totally agree with the reviewer that the cohort is imbalanced which limits conclusions of this study. However, this imbalance resulted from the entity / disease we assessed in the study. There are many hyperdense cysts seen in CT images every day; however, only a small part of them are vascularized lesions. Most vascular renal lesions present with solid components, wall thickening or contrast-enhancing nodules. This study excluded all cases with solid components, wall thickening or contrast-enhancing nodules, which are quite obvious to be malignant. We only included cases that were not obviously malignant; hyperdense cystic lesions without any solid components. And out of these, malignant lesions are very, very rare in clinical routine. Moreover, it is important to recognize these lesions as this has a high clinical impact. We agree that this is a pilot study due to the low number of vascular lesions and that the results must be confirmed in larger studies. Due to the rareness of the disease, we suggest performing multi-center studies.

We also believe that the results are important in clinical routine despite the low number of cases. PCD-CT allows spectral differentiation in every scan, without higher radiation dose or specific default settings. Why not have a quick look at iodine maps in cases with hyperdense renal cysts? It only takes short time and might filter out vascular lesions that are not obvious to be malignant at first glance. And on the other hand: hyperdense renal cysts without contrast enhancement (as shown in iodine maps) would not require further examinations.

Moreover, vascular renal lesions can differ based on their histology, leading to markedly different contrast enhancement patterns. A typical example is given by clear cell vs papillary renal carcinomas - the latter can be much harder to detect due to their faint enhancement, whereas clear cell RCC are usually easily detected as avidly hypervascular masses. Hence, it is important to specify which histotype of renal cancer was put under the 'vascular lesions' umbrella, and eventually do a subclass analysis of different histotypes - unless one decides apriori to enroll e.g. ccRCCs only.

Thank you for this comment. We agree with the reviewer that consideration of the histology is very important as these entities differ regarding contrast enhancement. Due to the low number of cases, subgroup analyses would not be feasible in this study cohort. However, we suggest to consider these subgroup analyses for further (multi-center) studies.

We added the following sentences in the limitations section:

“Due to the low number of vascular lesions, no subgroup analyses regarding the histopathology (e.g. clear-cell vs. papillary renal cell carcinoma) were feasible. However, these might present with different contrast enhancement and might be considered in larger study cohorts.”

Finally, it would be important to have a control group of patients imaged with a non-PCCT scanner (ideally a DECT, EID-based type) to seek differences in diagnostic performance. The advantages of DECT over single energy CT are well known since the early days of abdominal DECT, and comparing the study findings with non-PCCT DECT instead of presenting them without an external comparison would add considerable value to the manuscript.

Thank you also for this valuable comment. This is a great idea. Unfortunately we cannot perform this comparison as we do not have a cohort with available EID-based DECT examinations for the abdomen. We thought about including a non-DECT-based cohort as comparison; however, in this cohort we could only compare 70 keV images which we know that they are similar. Comparison of iodine maps and VNC would be very interesting, but not feasible in our cohorts. We added a suggestion for further studies in the discussion section:

“Moreover, a comparison between the diagnostic accuracy of DECT and PCD-CT for hyperdense cystic renal lesions would be of high importance”.

Reviewer 3 Report

Comments and Suggestions for Authors

Thanks for the opportunity to review the manuscript entitled “Spectral differentiation of hyperdense non-vascular and vascular renal lesions without solid components in contrast-enhanced Photon-Counting Detector CT scans – a pilot study “. The authors assessed the value of PCD-CT-derived material decomposition images to characterize renal cystic lesions as vascular or non-vascular. The topic is timely and of interest, and despite many strengths, the study also has potential drawbacks that should be addressed.

-          L60: this is exactly the same as for DECT. Please emphasize what is different with PCDs

-          The last sentence of the introduction defines the study aims; however, there is no gold standard mentioned. Please revise

-          L96, L130: “Renal cell carcinomas were classified as vascular renal lesions, and hemorrhagic or proteinaceous renal cysts were classified as non-vascular renal lesions.” This is fine, but please define more precisely the study process. Where the lesions classified before or after quantitative PCCT analysis? The reference standard is highly heterogeneous (5 different potential criteria applied in a way not clearly described), and with a small sample size the risk of bias is substantial.

-          Given the low level of standardization of the gold standard, I would suggest that table 1 should indicate what criteria were used to classify the 4 vascular and the 31 non-vascular lesions

-          Figure 2: please revise the labels; 0, 1, “non-vascular lesion » are confusing

-          Figure3: data presented is not complete. For the first case, I would like to see the VNC (including CT number measurement) and iodine concentration image (including concentration measurement). The same applies to the 2nd case.

-          L237: an iodine quantification map measures iodine concentration. Indicating HU does not seem appropriate or is at least unusual. Please elaborate on this. The same comment applies to iodine map measurements presented in table 3, and on L281.

-          L311: material decomposition is not unique to PCCT and is also possible with spectral EID-CT, especially with dual-layer CT. Please revise

-          L323: dual-layer CT also acquires spectral data routinely. Additionally, many centers using fast switching and dual-source DECT also routinely acquire spectral data on abdominal CT for the exact reason of characterizing incidental lesions, and thus have the spectral data available. This capability is not unique to PDC-CT.

-          L351: I am not sure to agree with this statement, many previous papers have used iodine concentration to differentiate cystic renal lesions.. Please take a look at the following papers, some of which you have cited. Can you elaborate on the advantage of reporting iodine vs. water material decomposition maps in HU, which seems counter-intuitive to me?

https://www.ncbi.nlm.nih.gov/pmc/articles/PMC6724630/

https://ajronline.org/doi/full/10.2214/AJR.18.20574

https://www.mdpi.com/2075-4426/13/11/1546

-          L352: based on my previous comment, please explain the physical relationship allowing to report an iodine concentration in HU. HU does not correspond to a concentration. Also, how could your results be applicable to other vendors in the future?

-          In the limitations, I agree that larger studies would be valuable, but also studies that are either (a) prospective or (b) retrospective with a unique, standardized gold standard for all patients.

Author Response

Spectral differentiation of hyperdense non-vascular and vascular renal lesions without solid components in contrast-enhanced Photon-Counting Detector CT scans – a pilot study (diagnostics-2774666)

We would like to thank the editor-in-chief and the reviewers for their effort and highly appreciate the constructive character of their criticism and comments. The manuscript was revised as suggested and resubmitted with the following point by point response to the comments of the reviewers. We believe that the quality of our manuscript substantially improved during the review process.

Response to Reviewer 3:

Thanks for the opportunity to review the manuscript entitled “Spectral differentiation of hyperdense non-vascular and vascular renal lesions without solid components in contrast-enhanced Photon-Counting Detector CT scans – a pilot study “. The authors assessed the value of PCD-CT-derived material decomposition images to characterize renal cystic lesions as vascular or non-vascular. The topic is timely and of interest, and despite many strengths, the study also has potential drawbacks that should be addressed.

Thank you very much for considering our manuscript and providing valuable comments.

-          L60: this is exactly the same as for DECT. Please emphasize what is different with PCDs

Thank you for this comment. We agree with the reviewer that further clarification is important; we added the following sentence:

“In contrast to DECT, PCD-CT offers spectral differentiation within each scan, without higher radiation dose or special protocols.”

-          The last sentence of the introduction defines the study aims; however, there is no gold standard mentioned. Please revise

Thank you for this suggestion; we added the reference standard in the end of the introduction section:

“Diagnostic workup for discriminating hyperdense renal cysts can be performed using either multiphasic CT, contrast-enhanced ultrasound (CEUS), MRI, PET-CT or histo-pathological diagnosis.  ”. 

-          L96, L130: “Renal cell carcinomas were classified as vascular renal lesions, and hemorrhagic or proteinaceous renal cysts were classified as non-vascular renal lesions.” This is fine, but please define more precisely the study process. Where the lesions classified before or after quantitative PCCT analysis? The reference standard is highly heterogeneous (5 different potential criteria applied in a way not clearly described), and with a small sample size the risk of bias is substantial.

Thank you for this comment. We are aware that the reference standard used in this study is heterogeneous. However, due to the retrospective study design it was not feasible otherwise. The lesions were classified after PCD-CT using the reference standards. However, this was performed within in clinical routine and not within the study; especially the quantitative measurements did not influence diagnostic work-up as these were performed retrospectively and diagnostic work-up was already performed in all cases. We believe that further, maybe also prospective, studies are important to reduce this bias and to build a homogeneous cohort with a comparable reference standard.

-          Given the low level of standardization of the gold standard, I would suggest that table 1 should indicate what criteria were used to classify the 4 vascular and the 31 non-vascular lesions

Thank you for this suggestion; we added the reference standard in Table 1.

-          Figure 2: please revise the labels; 0, 1, “non-vascular lesion » are confusing

Thank you, this was changed accordingly.

-          Figure3: data presented is not complete. For the first case, I would like to see the VNC (including CT number measurement) and iodine concentration image (including concentration measurement). The same applies to the 2nd case.

We thank the reviewer for this valuable advice. Figure 3 was changed accordingly and we added VNC and iodine maps for both cases.

-          L237: an iodine quantification map measures iodine concentration. Indicating HU does not seem appropriate or is at least unusual. Please elaborate on this. The same comment applies to iodine map measurements presented in table 3, and on L281.

We thank Reviewer 3 for this comment. The iodine maps used in this study represent series that show the proportional attenuation caused by iodine. Therefore, the units are in fact Hounsfield units. Ultimately, the iodine map represents what is subtracted from the contrast-enhanced image to obtain the virtual non-contrast image.

-          L311: material decomposition is not unique to PCCT and is also possible with spectral EID-CT, especially with dual-layer CT. Please revise

We agree with the reviewer that this sentence was quite confusing. We added the value of DECT and pointed out the advantages of PCD-CT.

-          L323: dual-layer CT also acquires spectral data routinely. Additionally, many centers using fast switching and dual-source DECT also routinely acquire spectral data on abdominal CT for the exact reason of characterizing incidental lesions, and thus have the spectral data available. This capability is not unique to PDC-CT.

Thank you for this comment. We added this sentence in the discussion to highlight the possibilities of DECT. At our institution, this method is not possible.

“However, many centers using fast switching and dual-source DECT also routinely ac-quire spectral data on abdominal CT for characterizing incidental lesions, and thus have the spectral data available.”

-          L351: I am not sure to agree with this statement, many previous papers have used iodine concentration to differentiate cystic renal lesions. Please take a look at the following papers, some of which you have cited. Can you elaborate on the advantage of reporting iodine vs. water material decomposition maps in HU, which seems counter-intuitive to me?

https://www.ncbi.nlm.nih.gov/pmc/articles/PMC6724630/

https://ajronline.org/doi/full/10.2214/AJR.18.20574

https://www.mdpi.com/2075-4426/13/11/1546https://www.mdpi.com/2075-4426/13/11/1546

Thank you for this comment and the suggestion to include these interesting studies in our discussion section. We added further information regarding this issue in the discussion section and cited these studies. 

-          L352: based on my previous comment, please explain the physical relationship allowing to report an iodine concentration in HU. HU does not correspond to a concentration. Also, how could your results be applicable to other vendors in the future?

We agree with Reviewer 3 that iodine concentration cannot be reported in Hounsfield units. However, in the present study we used iodine maps. These iodine maps represent the voxel-based attenuation of iodine measured in Hounsfield units. The VNC algorithm differentiates the iodine contrast and subtracts it from the contrast-enhanced image to obtain the virtual non-contrast image. The voxel-wise attenuation that is caused by iodine contrast then is used to generate the iodine map. Therefore, we reported the iodine attenuation in Hounsfield units.

-          In the limitations, I agree that larger studies would be valuable, but also studies that are either (a) prospective or (b) retrospective with a unique, standardized gold standard for all patients.

Thank you for this advice; we added this in the limitations section.

Round 2

Reviewer 2 Report

Comments and Suggestions for Authors

Thank you for your kind reply.

Comments on the Quality of English Language

The English language needs some minor editing.

Author Response

Thank you very much for your valuable comments and consideration of our manuscript. 

We performed further English language editing. 

Reviewer 3 Report

Comments and Suggestions for Authors

I thank the authors for adressing my comments.

Author Response

Thank you very much for your valuable comments and consideration of our manuscript.